# A High-Voltage and Low-Noise Power Amplifier for Driving Piezoelectric Stack Actuators

**DOI:** 10.3390/s20226528

**Published:** 2020-11-15

**Authors:** Lisong Xu, Hongwen Li, Pengzhi Li, Chuan Ge

**Affiliations:** 1University of Chinese Academy of Sciences, Beijing 100049, China; gechmail@163.com; 2Changchun Institute of Optics, Fine Mechanics and Physics, Chinese Academy of Science, Changchun 130033, China; lihongwen@ciomp.ac.cn; 3School of Computing and Engineering, University of Gloucestershire, The Park, Cheltenham GL502RH, UK; pli2@glos.ac.uk

**Keywords:** piezoelectric stack actuator, high-voltage operational amplifier, power amplifier, low output voltage ripple, precision positioning

## Abstract

In this paper, based on the principles of general operational amplifiers, a high-voltage operational amplifier is developed. Considering the influences of piezoelectric stack actuators on the circuit, a novel structure using the high-voltage operational amplifier as a noninverting amplifier is proposed. Because of the simple circuit principles and the voltage feedback control structure, the proposed power amplifier has the advantages of low noise and small size, and it can be realized by discrete electric elements easily. In the application of precision positioning, a power amplifier using the proposed circuit principles for driving piezoelectric stack actuators is designed, simulated, and tested. The simulated results show that the proposed power amplifier could conform to the theory of the circuit. The experimental results show that the designed power amplifier conforms to the simulation, the bandwidth of the power amplifier is about 57 kHz, and the ripple of the power amplifier is less than 2 mV. Furthermore, the output of the proposed power amplifier maintains the same type of wave within in a large range of frequency, while the input is the sinusoidal or square wave, and the resolution of the mechanism which the power amplifier is applied in is about 4.5 nm. By selecting the critical electronic elements and using feedback control, the proposed circuit structure is able to realize a low-cost and high-performance power amplifier to drive piezoelectric stack actuators flexibly, which is the novel work of the paper.

## 1. Introduction

Piezoelectric stack actuators have been widely used in robotics, scanning probe microscopies, industrial precision positioning systems, and biomedical engineering because of their small size, high displacement resolution, and lack of electromagnetic interference [1,2,3,4,5,6,7,8]. In these applications, power amplifiers are important subsystems, and low-noise and high-voltage power amplifiers are needed to achieve the performance of piezoelectric stack actuators. There are usually two types of power amplifiers: voltage amplifiers and charge amplifiers [9,10,11,12,13,14]. Because voltage amplifiers have the advantages of simple structure and high bandwidth, they have been widely adopted for commercial power amplifiers [13,15]. Many such voltage amplifiers have been proposed by References [16,17,18,19,20,21,22,23], and the power amplifier can also be used to estimate the force and displacement of the piezoelectric stack actuator [24]. Reference [25] studies and implements a 15 W driver for piezoelectric stack actuators; the power losses of the driver can be reduced because of the proposed circuit structure, and the output voltage ripples are about 40 mV. Reference [13] proposes a power amplifier based on the error-amplified principle, and the output voltage ripples are about 20 mV. In Reference [26], a dual amplifier able to deliver a 300 kHz sine wave of 20 Vpp amplitude with 100 nF load is proposed. Aiming at low cost, Reference [15] designs the power amplifiers by audio power amplifiers, because of their high output current and high frequency bandwidth, and the power amplifiers can drive capacitive loads with the output signal frequencies reaching 20 kHz. Reference [27] implements an innovative amplifier, using isolation amplifier and subtracting amplifier; the amplifier presents a voltage gain of approximately 100 and an output swing of 200 V. These works in the mentioned referenced papers are innovative; however, some problems still exist: high output noise ripples, high price, low output voltage, or complex circuit structure. Reference [13] shows that the core components of power amplifiers are high-voltage operational amplifiers with high current and low output noise. At present, there are commercial products from Apex Microtechnology (Tucson, Arizona, USA) [28], but they have disadvantages of high price and little choice; it could not satisfy some applications in which the amplifiers should be low in cost.

The structure and the principles of power amplifiers of low cost, small size, simple structure, and high precision are proposed in this paper. The proposed power amplifier is mainly based on the structure of the general operational amplifiers, and it takes the circuit model of piezoelectric stack actuators into account. The power amplifier could be realized by discrete electronic elements flexibly. Then a prototype power amplifier is produced. The performance of the developed prototype power amplifier is simulated by Cadence/PSPICE (San Jose, CA, USA) software and tested by a series of experiments. The novelty of the paper is that the proposed circuit structure is simple, and the function of each part is clear, which makes the realization and the test of the circuit easier. Furthermore, the circuit is realized by discrete electronic components, which makes it easy to adjust the circuit parameters to drive various piezoelectric stack actuators.

The organization of this paper is as follows. Section 1 gives a brief introduction. In Section 2, the structure and the principle of the power amplifier are analyzed explicitly; they reveal the feasibility of the design profoundly and show the design is applicable. Section 3 selects the crucial electronic components. Section 4 shows the simulation results of the power amplifier. Section 5 presents the experimental results of the developed power amplifier. Finally, Section 6 draws the main conclusions.

## 2. The Circuit Principles of the Power Amplifier

### 2.1. The Schematic of the Proposed High-Voltage Operational Amplifier

Based on the principles of general operational amplifiers, the schematic of the high-voltage operational amplifier is shown as Figure 1. According to Figure 1, the high-voltage operational amplifier is composed by three stages.


**The input stage**


The input stage of the high-voltage operational amplifier is a differential amplifier circuit. The circuit is composed by *Q*
_2–5_, *D*_1–3_ and some resistances, and the pair transistor *Q*_4_ is the critical element of the circuit. The pair transistor and *Q*_2–3_ realize a CC-CE combined amplifier; *R*_1_ and *D*_3_ realize a voltage clamp circuit, which makes the voltage between base of *Q*_2_ and emitter of *Q*_4_ constant. The current source of the differential amplifier circuit realized by *Q*_5_ and *R*_9_ ensures the symmetry of the circuit and provides an appropriate operating point for stage 2. The relationships between the input and output can be expressed as follows:(1)$$V_{1}(t)=V_{Q}+A_{1}(V_{i+}(t)-V_{i-}(t)),$$
where *A*_1_ is the magnification of the circuit, *V_Q_* is the output bias, and *V*_1_*(t)* is the voltage signal described in the Figure 1. *V_Q_* can be expressed as Equation (2), and the needed operation point *V_Q_*_2_ for stage 2 can be expressed by Equation (3). In Equations (2) and (3), *V_BE_* is the drop voltage of the diode and *VCC* is the voltage of the positive power supply.
(2)$$V_{Q}=VCC-(0.5I_{b}R_{3}+V_{BE}),$$
(3)$$V_{Q2}=VCC-(I_{a}R_{2}+V_{BE}).$$

Making *I_a_ = I_b_* and *R*_3_
*= 2R*_2_, and then *V_Q_ = V_Q_*_2_, the designed parameters can satisfy the requirement of the operational point of the stage 2 perfectly, which is the important role of the stage 1. The CC-CE combined amplifier has excellent frequency characteristics, and the magnification *A*_1_ is a constant value during DC to thousands Hz.


**The amplify stage**


Stage 2 of the circuit is configured as a common-emitter BJT with a current source load. The circuit always needs a stable operating point, which demands that the input signal is very small and has no effect on the operating point. The input voltage *V*_1_*(t)* described in Figure 1 can be expressed as follows:(4)$$V_{1}(t)=V_{Q2}+V_{S}(t),$$
where *V_Q_*_2_ is the operation point and *V_S_(t)* is the input signal. Based on the circuit theory, it is proved that the circuit only amplifies the input signal, and the relationship between the input and output signal can be expressed by Equation (5), where *A*_2_ is the magnification of the circuit and *V*_2_*(t)* is the voltage signal described in Figure 1.
(5)$$V_{2}(t)=A_{2}V_{S}(t).$$

An ideal current source has infinite internal resistance, but the characteristic of the current source realized by depletion MOSFETs is shown as Figure 2. During the operating range, the actual current source has a dynamic internal resistance, which is expressed in Figure 2.

Taking the model of the BJT into account, the model of the circuit within the bandwidth is shown as Figure 3. Ignoring *R*_5_ and *C*_1_, the magnification of the circuit is expressed as follows:(6)$$A_{2}(s)=-\frac{ri_{c}(s)}{i_{b}(s)R_{be}+(i_{b}(s)+i_{c}(s))R_{2}}=-\frac{r\beta }{R_{be}+(1+\beta )R_{2}}.$$

The dynamic internal resistance of the current source implemented by depletion MOSFETs is numerically big, so that the magnification of the circuit is also very big. The high-frequency attenuation is implemented by the network of *R*_5_ and *C*_1_, which improves the frequency characteristics of the circuit and enhances the stability of the circuit.


**The output stage**


The MOSFETs *M*_1_ and *M*_2_ are the critical elements of the output stage, which is a general class B amplifier circuit. In the circuit, *R*_7_ is used to eliminate crossover distortion; *R*_6_ and *R*_8_ are used to reduce the effect of the input capacitance of *M*_1_ and *M*_2_. The circuit has advantages of a large output current and high bandwidth, and its magnification can be expressed as *A*_3_*(s) ≈* 1.

When the power supply for the amplifier has the function of current protection or the output current of the amplifier is moderate, the circuit in Figure 1 can be applied directly. The practical circuit is shown in Figure 4 for when the demanded currents exceed the maximum currents of *M*_1_ and *M*_2_. As the parallel circuits of the class B amplifier are identical, the output current of the circuit can be expressed as Equation (7), which has enlarged the output current *N* times.
(7)$$\left\{ \begin{array}{l} I_{P1}=I_{P2}=\cdots =I_{PN}\\ I_{N1}=I_{N2}=\cdots =I_{NN} \end{array}\right.$$

Considering that the parameters of discrete electronic elements are inconsistent, the currents of *Mni* and *Mpi* need to be limited, so the MOSFETs are not damaged by transient current. In the circuit, the current of *Mpi* is limited by *Qpi* and *Rpi*, and the current of *Mni* is limited by *Qni* and *Rni*. The parallel connection of the class B amplifier is a major design method in the design of power amplifiers for piezoelectric stack actuators, which is different from a general operational amplifier, as the output current of a general operational amplifier is always about several mA.

In the actual model of MOSFETs, there are input capacitances, which are shown in Figure 5. As the number of the parallel connections of the class B amplifier increases, the bandwidth of the circuit reduces, which would reduce the stability of the high-voltage operational amplifier, and this is the cost of increasing the output current. Generally, the number of parallel circuits is determined by simulation.


**The overall magnification**


Based on the analysis of the three stages, the input–output relationship of the proposed high-voltage operational amplifier is described by the Equation (8), where *A* is the magnification of the proposed high-voltage operational amplifier and *A_1–3_* are the magnification of the three stages.
(8)$$V_{out}(t)=A_{1}A_{2}A_{3}(V_{in+}(t)-V_{in-}(t))=A(V_{in+}(t)-V_{in-}(t)).$$

### 2.2. The Principles Applying the High-Voltage Operational Amplifier as a Power Amplifier for Piezoelectric Stack Actuators

When applying the high-voltage operational amplifier to form a power amplifier, it is general to use feedback control, so that the magnification of power amplifiers is determined by the feedback network and the value of the magnification is constant during the designed frequency range. The feedback network is always formed by high-precision resistances and capacitances, which reduces the influence induced by discrete electronic elements of the high-voltage operational amplifiers. There are two methods to use the proposed high-voltage operational amplifier for piezoelectric stack actuator: the noninverting amplifier and the inverting amplifier. The noninverting amplifier used as a power amplifier for piezoelectric stack actuator is shown in Figure 6; it can isolate the input and output well. The transfer function of the circuit can be expressed as follows:(9)$$F(s)=\frac{A(s)}{1+A(s)H(s)}.$$

In Equation (9), *A(s)* is the transfer function of the operational amplifier, and *H(s)* is the transfer function of the feedback network, which is always realized by resistances and capacitances. As long as *A(s)H(s)* >> 1, Equation (9) can be expressed as follows:(10)$$F(s)\approx H^{-1}(s).$$

The stability of the power amplifier using feedback control needs to be analyzed and verified by simulations, especially when piezoelectric stack actuators are used as the load.

Considering the noise of circuit as disturbances, the model of the amplifier is shown as Figure 7. In the figure, *N*_1–3_*(s)* are the noise of the input stage, the amplifier stage, and the output stage, respectively, and the influences of the noise can be expressed by Equation (11). According to the equation, the influences of *N*_2_*(s)* and *N*_3_*(s)* would be attenuated as *A*_1_*(s) >>* 1 and *A*_2_*(s) >>* 1, so that only the noise of the input stage decides the output noise of the circuit. The input stage is composed by low-noise discrete elements, making the output noise of the circuit very low.
(11)$$\begin{array}{ll} Y_{N}(s) & =\frac{A(s)N_{1}(s)}{1+A(s)H(s)}+\frac{A(s)N_{2}(s)A_{1}^{-1}(s)}{1+A(s)H(s)}+\frac{A(s)N_{3}(s)A_{1}^{-1}(s)A_{2}^{-1}(s)}{1+A(s)H(s)}\\ & \approx H^{-1}(s)N_{1}(s)+H^{-1}(s)A_{1}^{-1}(s)N_{2}(s)+H^{-1}(s)A_{1}^{-1}(s)A_{2}^{-1}(s)N_{3}(s) \end{array}$$

### 2.3. The Current Requirements of the Power Amplifiers

Piezoelectric stack actuators have complex electrical and mechanical models [29], and the models have influences on the power amplifier. In general, the model of piezoelectric stack actuators is a capacitor in series with a resistor, which makes the power amplifier need the ability of high output current. In Reference [13], the model of piezoelectric stack actuators is proposed, as shown in Figure 8. As demonstrated in the figure, *Cx* is the capacitance value of piezoelectric stack actuators, and *Rx* is the dynamic power loss of piezoelectric stack actuators. The value of *Cx* is almost several µF, and the value of *Rx* is several Ohm, and these values can be acquired by impedance analyzer or get from the manufacturer. Taking the model of piezoelectric stack actuators into account, the whole circuit model of the power amplifier and piezoelectric stack actuators is shown in Figure 9. The current *I(s)* can be expressed as Equation (12):(12)$$I(s)=\frac{V(s)}{R_{s}+R_{x}+L_{x}s+\frac{1}{C_{x}s}},$$
where *Rs* is the output resistor of the power amplifier, the value is always less than 1 Ω, *F(s)* is the magnification of the power amplifier, and *Lx* is the inductance of the wire. The value of *Lx* is generally very small, so that its influence can be neglected. Then the output current *I(s)* can be expressed as follows:(13)$$I(s)\approx \frac{V(s)C_{x}s}{(R_{x}+R_{s})C_{x}s+1}.$$

When the input signal is explicit, the current of the power amplifier can be calculated by Equation (13), which determines the selection of the electronic elements and the parallel quantity of the current amplification stage of the power amplifier. Periodic sinusoidal tracking is common in the application of piezoelectric stack actuators. In that situation, the out current is calculated easily; let *s = wj*, and Equation (13) can be expressed as follows:(14)$$I(jw)=\frac{V(jw)C_{x}jw}{(R_{x}+R_{s})C_{x}jw+1}.$$

Let *R = R_x_ + R_s_* and the amplitude of *V(jw)* be *A*. When only amplitude is calculated, the Equation (14) is changed as follows:(15)$$\left|I(jw)\right|=\frac{C_{x}w}{\sqrt{R^{2}C_{x}^{2}w^{2}+1}}\left|V(jw)\right|=\frac{A}{\sqrt{R^{2}+\frac{1}{C_{x}^{2}w^{2}}}}.$$

According to Equation (15), the current is determined by the values of the *RC* and the input frequency when parameter *A* is determined. As the values of the *RC* are determined, the current increases by the increasing of the input frequency, and the maximal value of the current is *AR^−^*^1^. Let *w* = 2*πf* and *A =* 100; as the amplitude of output voltage is 100 V, the relationship between the input frequency and the current as the *RC* has a different value, as shown in Figure 10. According to Figure 10, the capacitance values of piezoelectric stack actuators and the maximum operating frequency are crucial parameter to design power amplifiers.

## 3. Setup of Prototype Circuit Board

In the theoretical analysis of Section 2, the design method of the power amplifier is proposed. Based on different type of piezoelectric stack actuators, it is easy to design the power amplifier by choosing appropriate electronic elements and applying feedback control. In a research of the projection lithography, the requirements are as follows:The piezoelectric stack actuator is P887.51 from Physik Instrumente (Karlsruhe, Germany) [30], whose electrical capacitance is about 3.1 µF.The output voltage range is −100 to +100 V, and the magnification is 16 times.The piezoelectric stack actuator works at an operating point; it is needed to track the input signal near the operating point, and the highest moving frequency of the mechanism is less than 100 Hz.The power amplifier, which should be easy to be integrated with the mechanism, must reserve negative voltage for other piezoelectric stack actuators from Physik Instrumente [30], such as P-143.05.

Based on the above requirements, the demanded output current is not large, as the piezoelectric stack actuators do not need to follow the input signal in the full range, so that, in stage 3, the supplied output current is composed by single MOSFETs, and the design could reduce the size of the power amplifier. Taking all the factors into account and referring to the circuit diagram in Figure 1, the results of the design are as follows:In the first and second stage, *Q*_4_ is *SSM2212*, *Q*_2–3_ are *CZTA44*, and *Q*_1_ is *FTZ560*, while the current source is realized by *CPC5603* that is *Q*_5_ or *Q*_6_ in Figure 1. The selection of *Q*_1–3_ and *Q*_5–6_ is based on the maximum Collector-Emitter Voltage or the maximum Drain-Source Voltage of these elements. *Q*_4_ is dual-matched NPN transistor whose input noise is low, and *SSM2212* matches it well.In the third stage, *M*_1_ and *M*_2_ are *IXFR15N100Q3* and *IXTH10P60P*, which both have more than 45 A instantaneous current and more than 10 A continuous current. The most powers of the MOSFETs are 400 and 460 W, respectively. The selection of *M*_1_ and *M*_2_ is mainly based on *V_DS_* and *I_D_*, which are the maximum Drain-Source Voltage and the maximum Drain Current, respectively, and the heat-dissipation performance of these elements should be taken into account.To achieve 16-times magnification, the structure of the noninverting amplifier is used, and the values of resistors are 10 and 150 k, respectively. In order to enhance the stability of the power amplifier, the resistor of 150 k is paralleled with a capacitor of 15 pF.

In order to confirm the effectiveness of design, the simulation of the power amplifier is done in the next section.

## 4. Simulations and Analysis

### 4.1. The Simulated Frequency Response of the Power Amplifier

In order to analyze the characteristic of the proposed power amplifier, the simulation of the power amplifier is implemented with the software Cadence 16.6. In the simulation, the positive and negative DC power supplies are set as +110 and −110 V, respectively.

Based on the circuit theory shown by Figure 6 in Section 2, it is convenient to form a power amplifier to drive piezoelectric stack actuators, and the simulated amplitude frequency response of the power amplifier is shown in Figure 11. According to the figure, the following conclusions could be drawn: The power amplifier is stable; the magnification of the power amplifier in the passband is 24.082 dB, which agrees with the theoretical magnification (16 times); the bandwidth is about 68 kHz, and the 0 dB frequency is about 3.5 MHz; and the magnification is flat during the passband of the power amplifier, which is suitable to form a high-precision and highly dynamic piezoelectric positioning system.

### 4.2. The Simulated Response of the Power Amplifier with Load

The simulated waveform of the power amplifier as sinusoidal input with piezoelectric stack actuators is to verify the ability to respond to the sinusoidal signal. In the simulation, a capacitor of 3.1 µF in series with a 10 Ω resistor is used to model the piezoelectric stack actuator, as the capacitor of the piezoelectric stack actuator (P887.51) is 3.1 µF. Using sinusoidal wave with amplitude of 6 V and frequency of 100 Hz as input, which conforms to the working conditions of the power amplifier, the simulated input and output waveforms are shown in Figure 12, the currents of the MOSFETs are shown in Figure 13, and the power consumptions of the MOSFETs are shown in Figure 14. According to the simulated results, the power amplifier has an excellent ability to respond to sinusoidal input, and the current and power consumption are in a safe work area of the MOSFETs.

Just as the above setup, the simulated waveforms of the power amplifier as square input are shown in Figure 15, Figure 16 and Figure 17. The input square wave has the amplitude of 6 V, the frequency of 100 Hz, and the rising time and failing time of 1 µs. The input and output waveforms of the simulation are shown in Figure 15, which contains the rising and failing edge details. The current of the MOSFETs is shown in Figure 16, and the power consumptions of the MOSFETs are shown in Figure 17. According to the simulated results, the power amplifier has an excellent ability to respond to step input, the most instantaneous currents of the MOSFETs are less than 15 A, and the most instantaneous power consumptions are about 1500 W. The instantaneous currents of the MOSFETs are less than the maximum instantaneous currents of the MOSFETs, which are about 40 A. The instantaneous power consumptions of the MOSFETs are about 1500 W, and the impulse time is about 200 µs. The average power of the MOSFETs in a period can be expressed by Equation (16): (16)$$P=\;P_{I}\frac{T_{I}}{T},$$
where *P* is the average power, *T* is the period, *P_I_* is the instantaneous power, and *T_I_* is the instantaneous time. In Equation (16), the instantaneous power is idealized as a narrow pulse. Based on Equation (16), the average power is about 30 W, as the period is 10 ms. The average power of the MOSFETs is less than the maximum powers of the MOSFETs, i.e., 400 and 460 W, respectively. However, the instantaneous power consumptions of the MOSFETs are so big that the MOSFETs should well dissipate heat to avoid damage.

The simulated results show that the designed power amplifier is consistent with the analysis in Section 2, and the power amplifier is stable and has excellent waveform response. The design results can be further verified by experiments.

## 5. Experimental Results and Comments

### 5.1. Experimental Setup

The test setup of the power amplifier is shown in the Figure 18. The main instruments are as follows: The power supply, which is customized from Chaoyang Power Supply Corporation, has the ability of current protection, whose maximum continuous current is 4 A and instantaneous current is 20 A, as the instantaneous time is 2 ms; the waveform generator is 33522B from Keysight Technologies (USA) [31]; the oscilloscopes measuring the waveforms is MSO5204 from Tektronix Corporation (USA) [32]; the multi-meter measuring DC voltage is 34470A form Keysight Technologies [31]; the network analyze testing the system frequency response characteristic is E5061B from Keysight Technologies [31]; and the piezoelectric stack actuator is P887.51 from Physik Instrumente [30].

### 5.2. Frequency–Amplitude Characteristic

The frequency–amplitude characteristic of the power amplifier is shown in Figure 19. According to the figure, the passband gain of the system is 24.026 dB, −3 dB bandwidth is about 57 kHz, and 0 dB bandwidth is about 2.388 MHz. The overall shape of the frequency–amplitude characteristics is consistent with the results of simulation analysis; there are a few differences between the tested frequency and the simulated frequency at the key frequency points, but the differences are not significant. The reasons for the differences are that the simulation uses an ideal electronic elements model which could not be found, and the simulation does not consider the distributed parameter of the print circuit board. As the differences are slight, especially in the low-frequency range, the simulation results can be used as references in the design.

### 5.3. The Ripple of the Power Amplifier

The out ripple is the important performance of various power amplifiers. As to the linear power supply, such as the designed power amplifier, the ripple is the output noise. The output ripple of the power amplifier would influence the output displacement accuracy of the piezoelectric stack actuator, especially in high-precision applications. Oscilloscope MSO5204 from Tektronix Corporation [32] was used for ripple test, with P2221 from Tektronix Corporation [32] as probe. We connected the input to ground, and the achieved output ripple is shown in Figure 20. According to the figure, the ripple of the power amplifier is less than 2 mV.

### 5.4. Tests with Piezoelectric Stack Actuator

In order to verify the stability of the power amplifier, a piezoelectric stack actuator (P887.51) is connected to the power amplifier. Using 100 Hz sinusoidal wave or 100 Hz square wave whose amplitude are both 6 V as input, respectively, the responses are shown in Figure 21 and Figure 22, which agree with the simulation results. The reason of choosing 100 Hz as the frequency point is that the highest moving frequency of the mechanism is 100 Hz.

### 5.5. Tests in the System

The mechanical stage for which the piezoelectric stacks actuator (P887.51) is applied is shown in Figure 23. The mechanical stage of 1 degree of freedom has a nominal displacement of 0–12 μm, which can be measured by Heidenhain LIP281 integrated in the stage. The stage is used to move required displacement and do some local adjustments at the position, and it is electrically excited by the designed power amplifier. The power amplifier uses the waveform generator (Keysight 33522B) to produce the input signal, and the power amplifier is powered by the customized power supply. The laboratory is under precise environmental control, with its ambient temperature kept at 22 ± 0.2 °C.

The noise of the used displacement sensor is shown in Figure 24a. According to the figure, the noise of the sensor is about 1 nm. The resolution of the mechanical stage is shown in Figure 24b, as the input signal is a square wave whose period and amplitude are 500 ms and 3 mV, respectively. According to Figure 24, the resolution of the mechanism is about 4.5 nm. The input–output relationship of the mechanical stage is shown in Figure 25; the step voltage and the step time of the input signal are 1.2 V and 200 ms, respectively. According to Figure 25, the output step displacement is about 2.5 μm, and the rise and fall characteristics are not consistent, due to the hysteresis effect of the piezoelectric stacks actuator. All of these results show that the designed power amplifier has excellent performances to drive the piezoelectric stacks actuator.

### 5.6. Comparisons

The comparisons between the proposed power amplifier and some existing power amplifier are listed in Table 1, and the comparisons are carried out as the power amplifies are without load. According to the table, the designed power amplifier has the advantages of low noise, small size, and low price. However, every coin has two sides; the design involves much work on the selection of the electronic elements, which increases the difficulty of design. As the paper discussed the principles of the circuit thoroughly in Section 2, the adverse factors in the design are minimized.

## 6. Conclusions

In this article, aiming at low cost, small size, and high precision, a power amplifier which conforms to the design principles of general operational amplifier is explored. Based on the theory of general operational amplifiers, the circuit structure of the power amplifier is proposed, and the method of applying the proposed amplifier to form a piezoelectric driver is described. The unique design requirements produced piezoelectric stack actuators, which are similar to the series connection of capacitance and resistance, were analyzed. In a precision positioning application, using P887.51 from Physik Instrumente [30], core electronic elements were selected. Using the PSPICE models of the elements, the designed circuit was simulated and confirmed to meet the requirements. Then, by testing a prototype circuit, experimental data and curves were achieved, and the experimental results are consistent with the simulation results perfectly. The experimental results demonstrate that the bandwidth of the power amplifier is about 57 kHz, the output ripple is less than 2 mV, the displacement resolution of the mechanical stage which the power amplifier is applied in is about 4.5 nm, and the proposed power amplifier can well driver the piezoelectric stack actuator during the travel range of the mechanical stage.

## Figures and Tables

**Figure 1 sensors-20-06528-f001:**
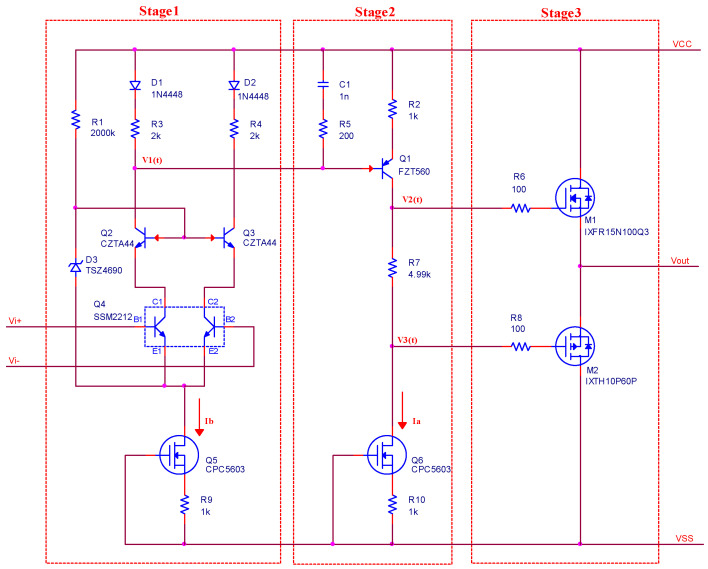
The schematic of the high-voltage operational amplifier.

**Figure 2 sensors-20-06528-f002:**
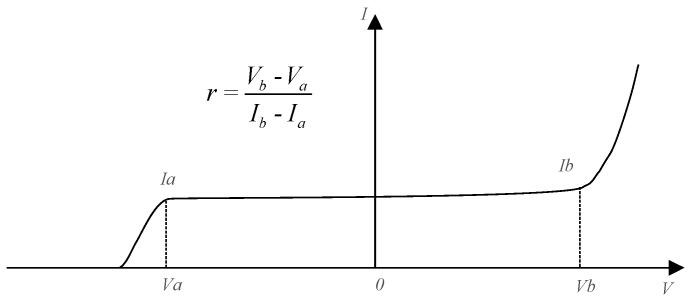
The dynamic resistance of the current source.

**Figure 3 sensors-20-06528-f003:**
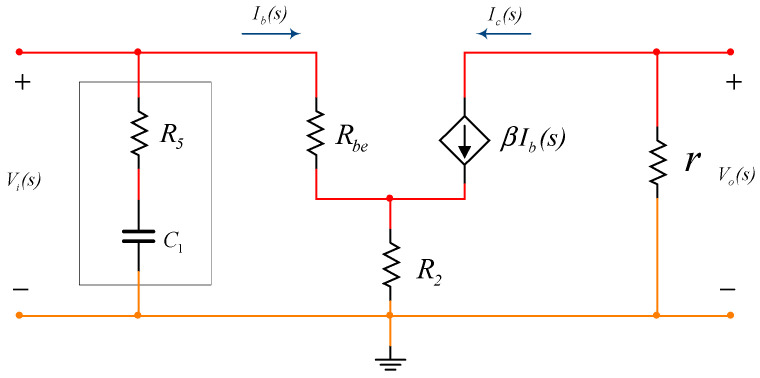
The equivalent circuit of the voltage-amplifying stage.

**Figure 4 sensors-20-06528-f004:**
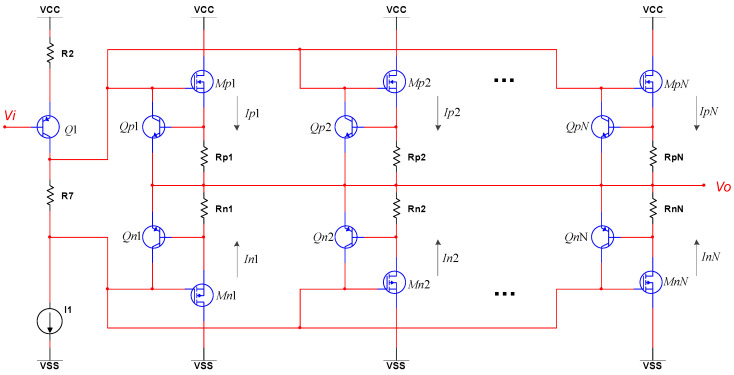
The power-amplifier circuit with multi-level parallel connection.

**Figure 5 sensors-20-06528-f005:**
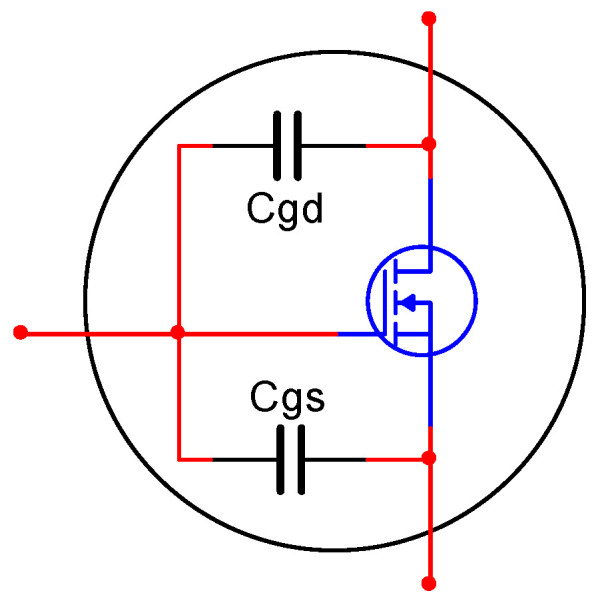
The input capacitance of MOSFETs.

**Figure 6 sensors-20-06528-f006:**
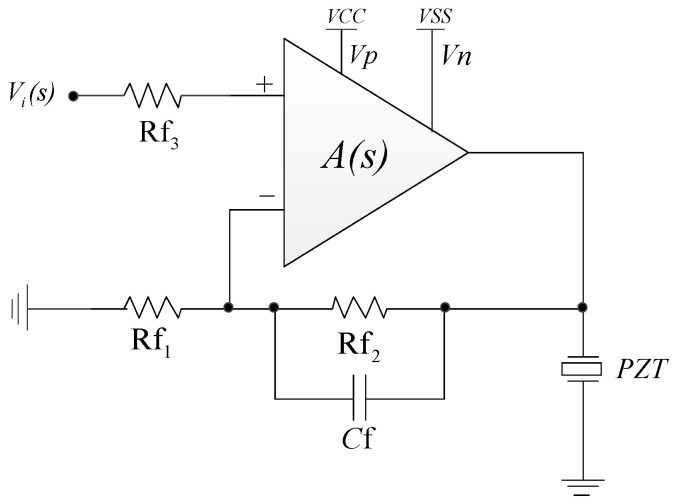
The noninverting amplifier is used as a power amplifier.

**Figure 7 sensors-20-06528-f007:**
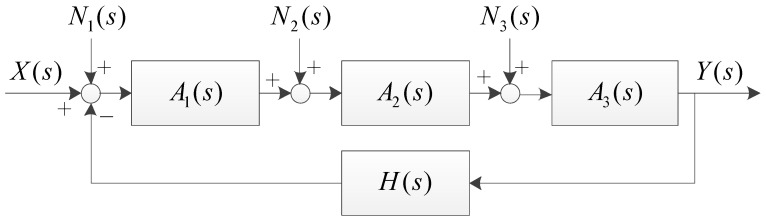
The noise model of the power amplifier.

**Figure 8 sensors-20-06528-f008:**
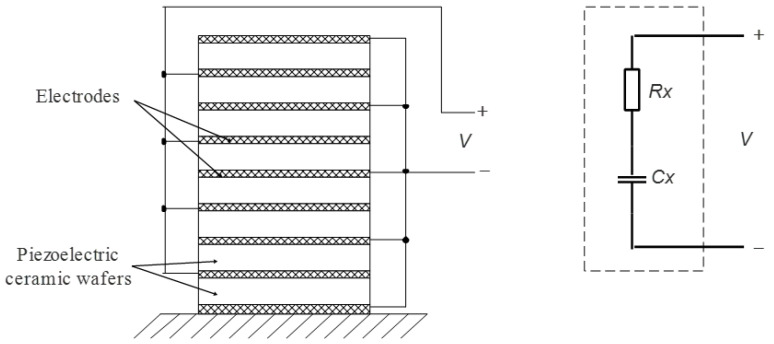
The model of piezoelectric stack actuators.

**Figure 9 sensors-20-06528-f009:**
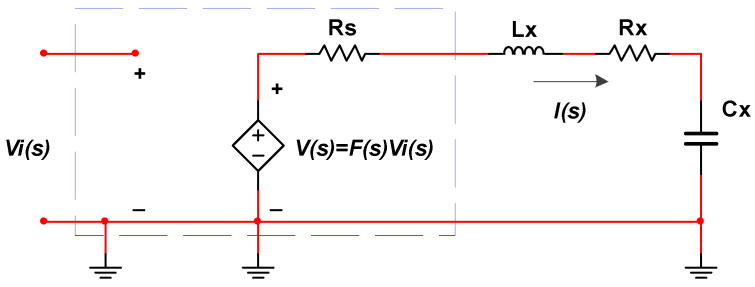
Frequency domain model of the circuit with piezoelectric stack actuators.

**Figure 10 sensors-20-06528-f010:**
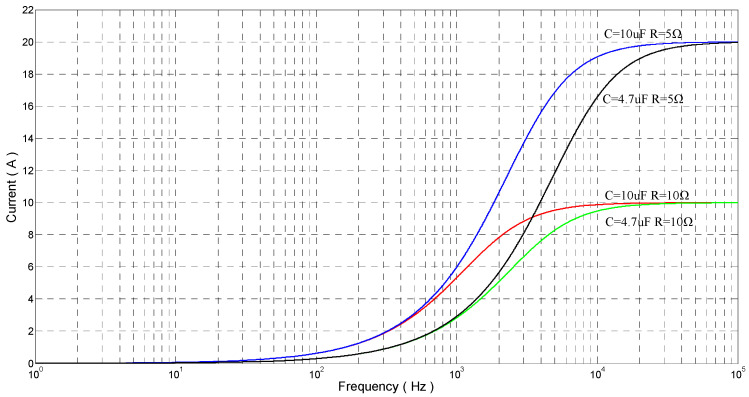
The relationship between current and frequency as the input is sinusoidal wave.

**Figure 11 sensors-20-06528-f011:**
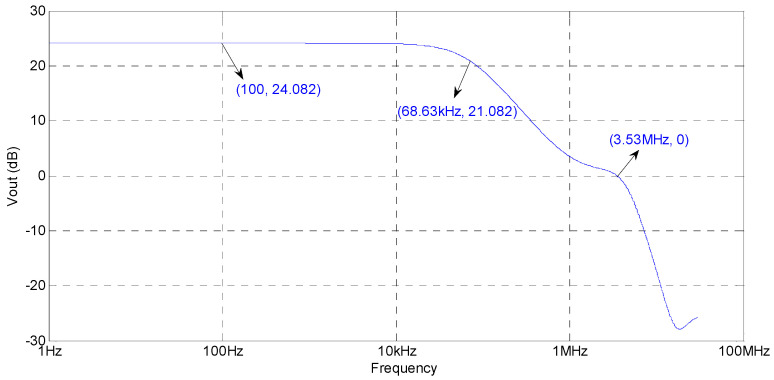
Simulated frequency response of the power amplifier.

**Figure 12 sensors-20-06528-f012:**
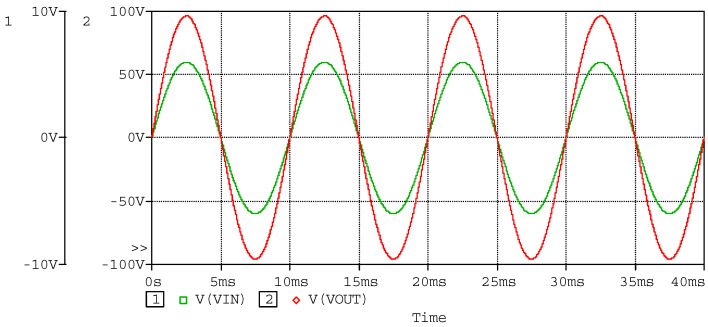
Simulated waveforms of input and output, as the input sinusoidal wave has amplitude of 6 V and frequency of 100 Hz.

**Figure 13 sensors-20-06528-f013:**
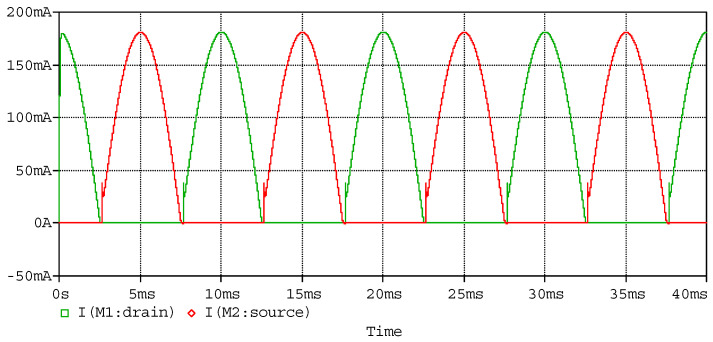
Simulated currents of MOSFETs.

**Figure 14 sensors-20-06528-f014:**
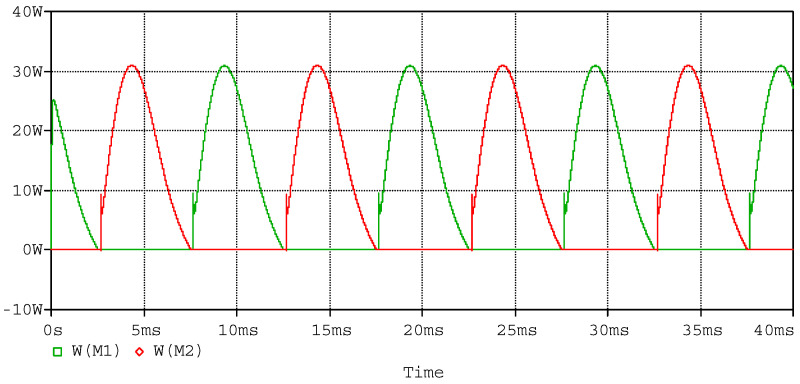
Simulated power consumptions of the MOSFETs.

**Figure 15 sensors-20-06528-f015:**
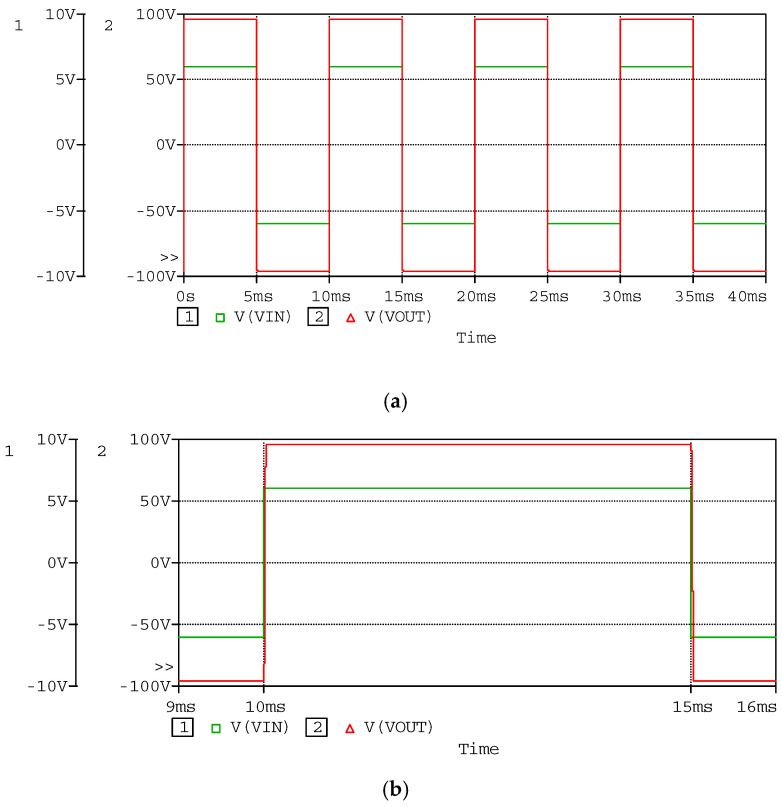
Simulated waveforms of input and output, as the input square wave has amplitude of 6 V and frequency of 100 Hz; (**a**) the overall perspective waveform; and (**b**) the rising edge and the falling edge of the waveforms.

**Figure 16 sensors-20-06528-f016:**
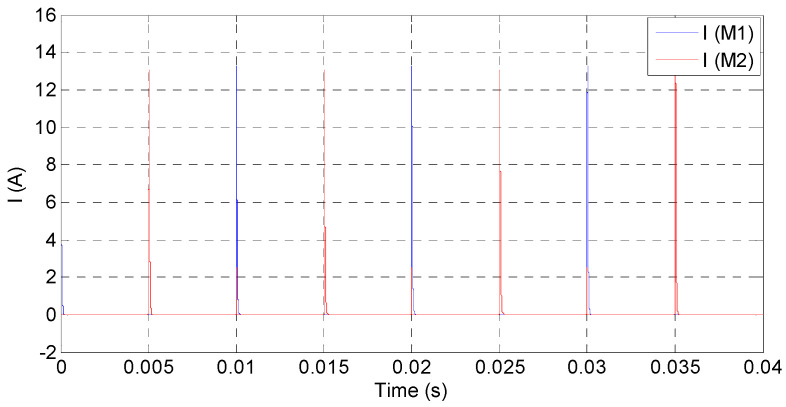
Simulated currents of MOSFETs.

**Figure 17 sensors-20-06528-f017:**
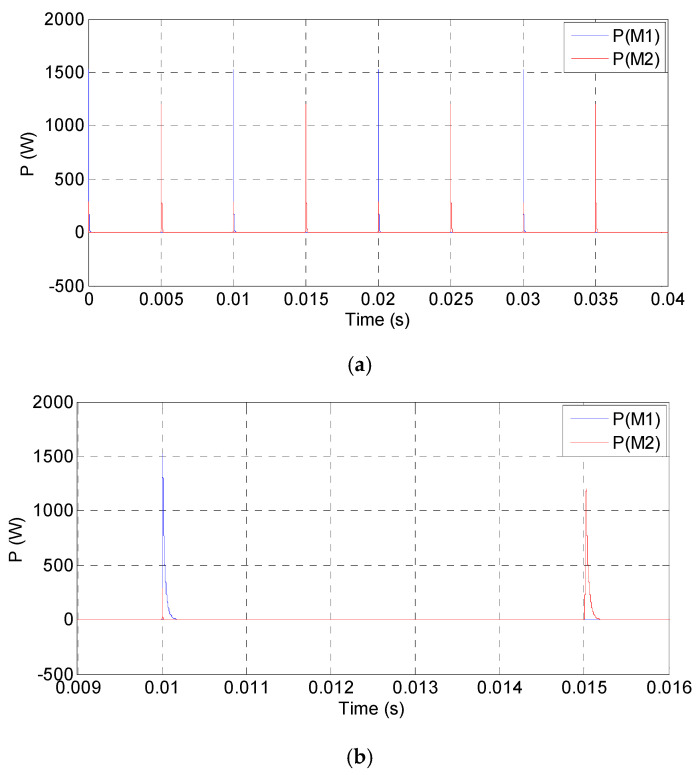
Simulated power consumptions of the MOSFETs: (**a**) the overall perspective waveform and (**b**) the detailed power consumptions of M_1_ and M_2._

**Figure 18 sensors-20-06528-f018:**
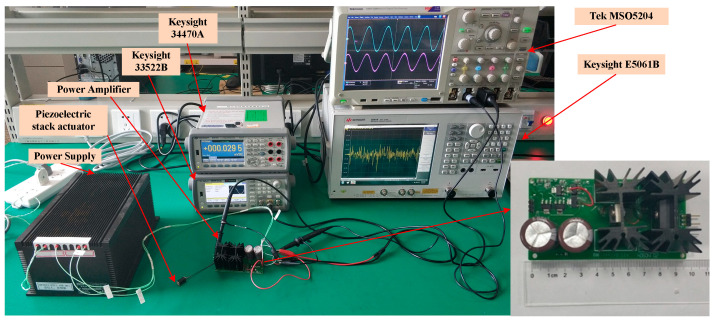
Photograph of the power amplifier’s test.

**Figure 19 sensors-20-06528-f019:**
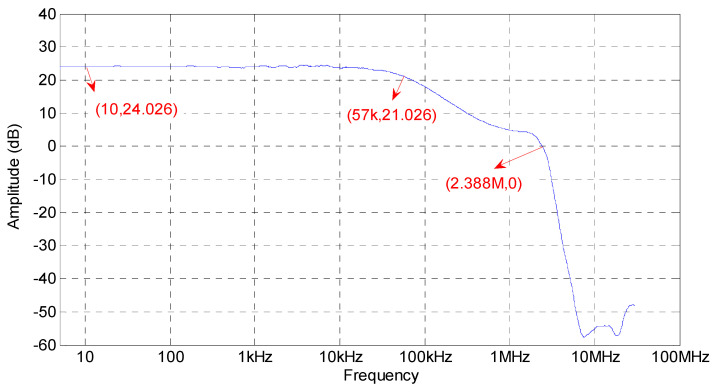
Measured frequency–amplitude characteristic.

**Figure 20 sensors-20-06528-f020:**
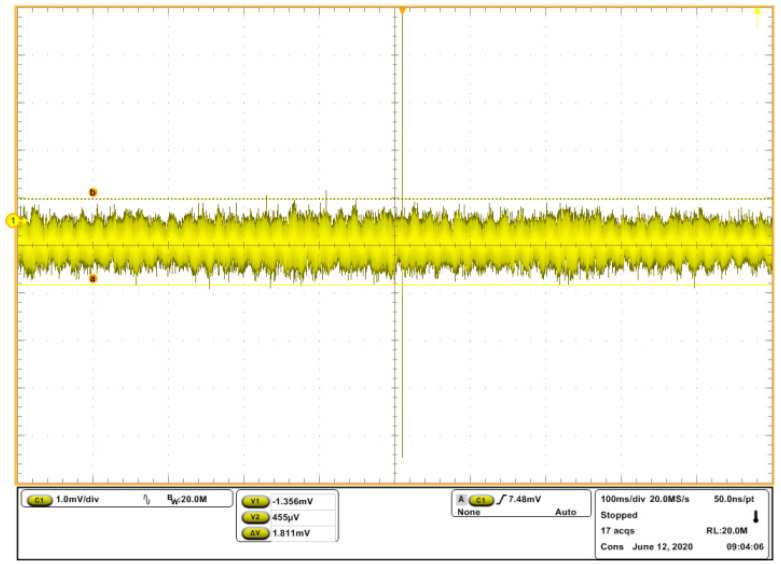
Measured ripple of the power amplifier.

**Figure 21 sensors-20-06528-f021:**
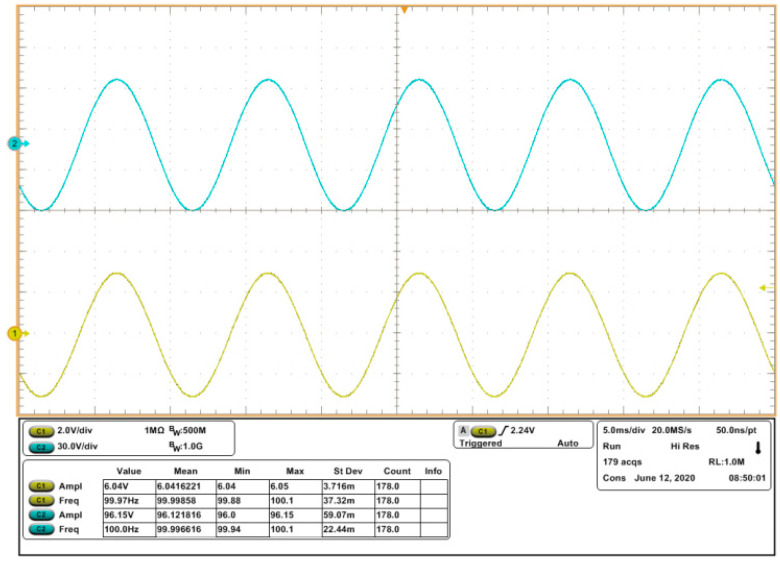
Measured waveforms of input (c1) and output (c2), as the input sinusoidal wave has an amplitude of 6 V and frequency of 100 Hz.

**Figure 22 sensors-20-06528-f022:**
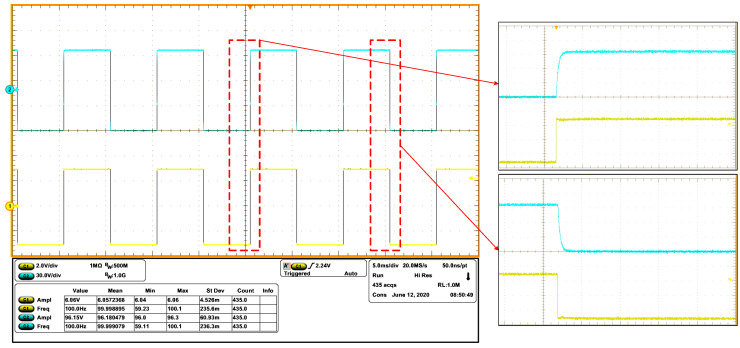
Measured waveforms of input (c1) and output (c2), as the input square wave has amplitude of 6 V and frequency of 100 Hz.

**Figure 23 sensors-20-06528-f023:**
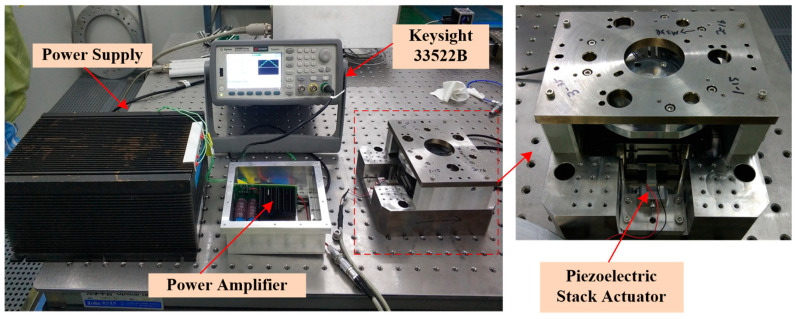
The setup of the tests in the system.

**Figure 24 sensors-20-06528-f024:**
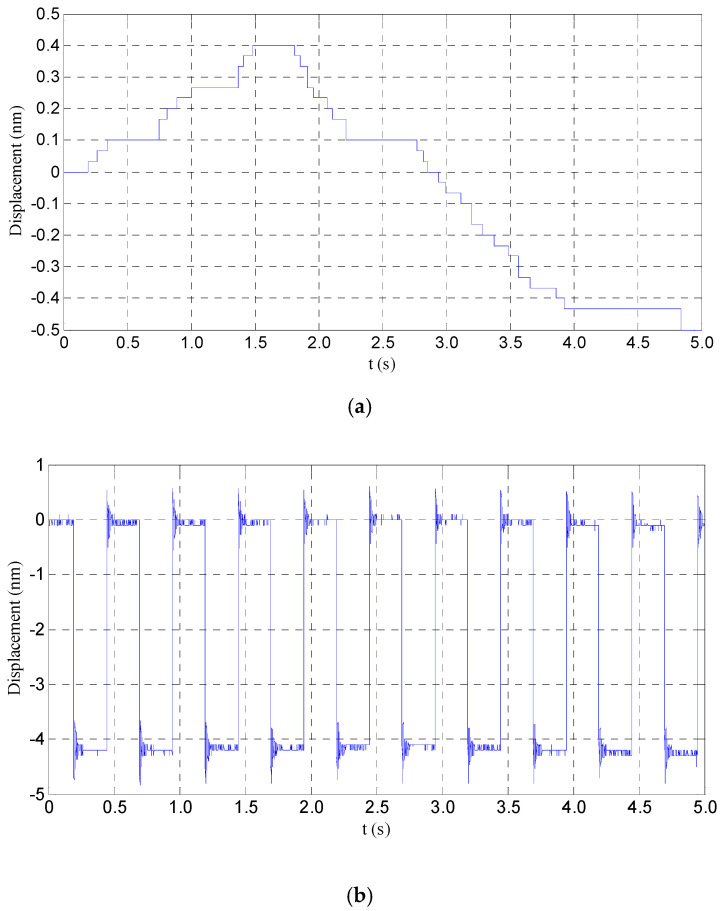
The output displacement resolution of the mechanism: (**a**) the noise of the sensor, and (**b**) the output displacement resolution of the mechanism.

**Figure 25 sensors-20-06528-f025:**
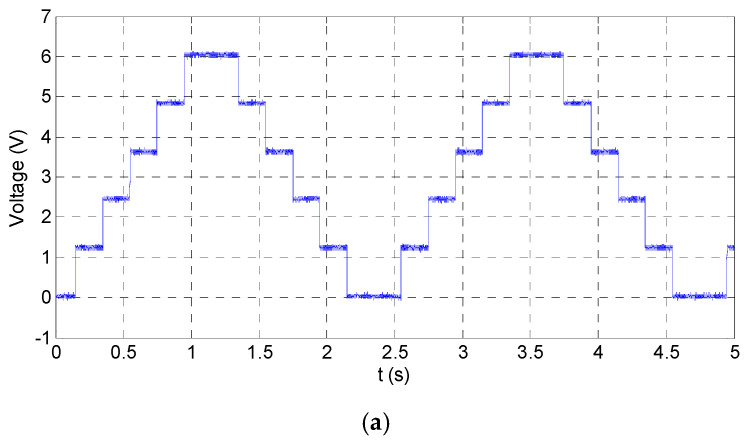
Measured input–output relationship of the mechanical stage: (**a**) the input voltage signal, and (**b**) the output displacement of the mechanical stage.

**Table 1 sensors-20-06528-t001:** Comparisons between the proposed power amplifier and some existing power amplifiers.

References	Ripple	Bandwidth	Price	Size
[13]	20 mV	About 20 kHz	Medium	Big
[15]	Not give	20 kHz	Medium	Medium
[25]	40 mV	Not give	Low	Medium
E-618 (Physik Instrumente [30])	20 mV	About 15 kHz	High	Big
This work	2 mV	57 kHz	Low	Small

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
