# Peer review of "A High-Voltage and Low-Noise Power Amplifier for Driving Piezoelectric Stack Actuators"

_sensors, 2020, doi:10.3390/s20226528_

Round 1

Reviewer 1 Report

In this paper, a power amplifier with high voltage and low noise was developed for driving piezoelectric stack actuators. A series of analyses including circuit principles of the power amplifier and PSPICE simulations were presented. Some experiments were performed to verify the theoretical design. This work is interesting, but several questions need to be illustrated for improving this paper.

  1. In lines 153, the author says that “it is natural that all these magnifications are the function of frequency”. But there is no distinct presentation (for example, formula) indicating the magnifications A1, A2 and A3 are related to the frequency in this paper. This point should be clarified.
  2. Please indicate the supplier and specific model of the DC power supply used in the experimental setup.
  3. 20 shows that the measured ripple of the developed power amplifier is less than 2 mV. However, it can be seen that the noise level of the output signal is obviously more than dozens of millivolt from Fig. 25, and the author claims that the developed power amplifier has advantage of low noise. These confuse the reader. Would the author can illustrate the relationship between the ripple and the noise?
  4. Actually, it’s difficult to obtain low ripple by using discrete components due to unavoidable changes of their characteristics and complex circuit structure. But the presented results in the paper are with advantages over previous works, which is significant. Perhaps some special methods have been used to eliminate noise. Would the author can add more discussions about electromagnetic compatibility design (grounding method, shield method etc.) into this paper? In another words, how to eliminate noise or reduce ripple of the power amplifier from aspects of design and prototype?
  5. In this paper, some experiments were carried out to verify the performances of the developed power amplifier. But the presented experimental results cannot fully demonstrate the designed performance of the power amplifier. Would the author can present the measured displacement of piezoelectric stack actuator excited by the developed amplifier with amplitude of +/-100V and frequency of 100Hz, which can illustrate the drive ability of the power amplifier more directly.
  6. 24 presents the excellent output displacement resolution (4.5nm) of the mechanism. It is suggested to add more illustration about the performances of the used displacement sensor into the paper (measurement resolution or accuracy).
  7. Although part of models of circuit component are presented in the paper. It is suggested to present the parameter of all components in Fig.1 (the used parameters of circuit components including resistors and capacitors).
  8. 10 shows the relationship between current and frequency under different value of RC components. But R has two parts of Rx (dynamic power loss of the piezoelectric stack actuator) and Rs (output resistor of the power amplifier). How to confirm Rx and Rs in design process? This point should be illustrated in the paper.

Reviewer 2 Report

 In this new submission, I have few comments.

1- Since the major contribution of this article is designing an operational amplifier using discrete components, a table of component list after section 3 would be appropriate. In my opinion, comparison table after section 5.6 is not complete and the parameters such as “Ripple” and “Bandwidth” are load dependent.

2- It seems that because of the reviewers` suggestions, the authors have revised references quite significantly. I have the opinion that; when giving general introduction, original articles should be cited at the beginning of the references. If the authors would like to cite a stick slip inertia motor for various applications of piezoelectric actuators, this should be an original article but not the recent ones as in [3] and [4]. For example, the one below has a publication date of 1990 (30 years ago).

Yamagata, T. Higuchi, H. Saeki, and H. Ishimaru, “Ultrahigh vacuum precise positioning device utilizing rapid deformations of piezoelectric elements”, Journal of Vacuum Science & Technology A 8, 4098 (1990); doi: 10.1116/1.576446.

I think the first 5 citations in the previous version are more appropriate.

Round 2

Reviewer 1 Report

The authors had revised this paper in a good way. I think it can be accepted for publication at present form.

This manuscript is a resubmission of an earlier submission. The following is a list of the peer review reports and author responses from that submission.

Round 1

Reviewer 1 Report

I am not sure if “design process” of a piezoelectric actuator driver (or amplifier) circuitry would be novel enough for a publication. Authors should emphasize novelty and present their contribution more clearly if there is any.

Introduction was written poorly. It only consists of sentences starting with “The paper…” or “in the paper”.

There are few round comments or statements like:

lines 60, 61: “some special application”, what are these special applications?

Line 79 : “some differences …” What are the differences, should be specified.

Line 220, 221: Based on different requirements…” What are these requirements?

Line 229 : “from different corporations”, which corporations?

In eqs (2) and (3), is it possible that Ia and Ib are changed?

On Fif. 1, input voltages are represented as Vi+ and Vi-, however in equations (1) and (4) , they are written as Vin+, Vin- and Vin, respectively. I think they should be consistent.

When preparing a setup to test the driver performance ( also in Fig 26) , a piezoelectric actuator was used as part of the experiment. When presenting results of the amplifier performance  in section 5.7, it was mentioned that the tests were performed with a 10µF capacitor and 10 ohm resistor. The title of the sub section is “Tests with piezoelectric stack actuator “, ??

Quality of figures from 12 – 24 and 28-35 are really poor so they are very difficult to follow due to poor quality. Curves and labels cannot be seen clearly or no labels’ at all.  Overall they should be improved significantly.

Reviewer 2 Report

Reviewer’s comments and suggestions for the Authors:

  1. The low cost, low noise, small size, and realization of discrete electric elements easily power amplifier by driving the piezoelectric stack actuators flexibly has been proposed in this research and explored at various capacitance and dynamic power loss value of piezoelectric stack actuators.
  2. Recently the piezoelectric stack actuator has been actively developed therefore in this paper the referring papers related in piezoelectric stack actuator after 2017 should been cited, as described below.

Title: Simultaneous Displacement and Force Estimation of Piezoelectric Stack Actuators Using Charge and Voltage Measurements

Authors: Sepehr Zarif Mansour ; Rudolf J. Seethaler

Journal: IEEE/ASME Transactions on Mechatronics,  Volume: 22(6), pp.2619-2624, 2017 

  1. The scheme of high voltage operational amplifier and its application in power amplifier have been described in this paper in section 2-1 and section 2-2 from Fig.1-Fig.7 in detail. However, the piezoelectric stack actuator is a key element in the power amplifier system therefore the piezoelectric stack actuators and its application should be described in this paper in detail and the related citations of the piezoelectric stack actuator should be also exhibited, as described below.

Title: Compact piezoelectric stacked actuators for high power applications

Authors: Kui Yao ; K. Uchino ; Yuan Xu ; Shuxiang Dong ; Leong Chew Lim

Journal: IEEE Transactions on Ultrasonics, Ferroelectrics, and Frequency Control, Volume: 47(4), pp.819-825, 2000

Title:A Novel Self-Sensing Stacking Piezoelectric Actuator Based on Structural Integration

Authors:Xin Dong ; Yuancai Yang ; Chong Zhang ; Xiaobiao Shan

Paper:2018 5th International Conference on Information Science and Control Engineering (ICISCE), 2018 

Title: Piezoelectric stack actuator parameter extraction with hysteresis compensation

Authors:Tiberiu-Gabriel Zsurzsan ; Charles Mangeot ; Michael A.E. Andersen ; Zhe Zhang ; Nils A. Andersen

Paper: 2014 16th European Conference on Power Electronics and Applications, 2014

Title:The research on a novel linear stepping piezoelectric stacks actuator

Authors:Hong-xuan Zhang ; Liang Wang ; Jia-mei Jin

Paper:2015 Symposium on Piezoelectricity, Acoustic Waves, and Device Applications (SPAWDA)

Reviewer 3 Report

see the attached pdf file

Round 2

Reviewer 1 Report

Language of the manuscript should be revised properly. Even if the authors mentioned that the introduction was revised, I do not see any improvement.

Rather than capturing the plots directly from the spice program, the data points should be saved as text and later plotted using another drawing program with proper curve and axis labeling.

The plots are occupying unnecessarily large area in the text that disturb the reading. Some curves should be combined. For example, the curves of (Figures 15B and C, 19B and C, 21B and C) can be plotted on the same plot.

The curves on Fig 25 is no longer readable.

All the captured plots from oscilloscope and network analyzer are annoying. As I have mentioned above, rather than capturing plots, the data points should be recorded and plotted properly using a drawing program. Here also some curves should be combined (Figures 30B and C, 32B and C).

There are also some omitting possibilities in the text. As the inductance (Lx) was omitted and s is replaced by w, Rs+Rx was combined as R and input is sinusoidal, then Eq (16) can easily come after eq. 12 without last term in eq (12) and middle term in eq. 16.

Authors should also decide the Oscilloscope type that they used; MOS5204 in line 373, MSO5204 on Fig. 23 and DPO5030 on line 406. It is a minor typo but decreases the credibility.  

Round 3

Reviewer 1 Report

I see some improvements in the manuscript.

In my opinion, quality of figures can still be improved. While reading a paper electronically, you stop and increase the enlargement rate to see a figure clearly is not a convenient way.  If figure style fits to the style format, I do not have any objection.

In figure 11, input vertical axis is from -10mV to +10mV. Is the output scale correct (from 107 to 108V)?
